# Leaving academia: PhD attrition and unhealthy research environments

**Andrea Kis[ID]\*, Elena Mas Tur, Daniël Lakens, Krist Vaesen[ID], Wybo Houkes**

School of Innovation Sciences, Eindhoven University of Technology, Eindhoven, The Netherlands

\* a.kis@tue.nl

## Abstract

This study investigates PhD candidates' (N = 391) perceptions about their research environment at a Dutch university in terms of the research climate, (un)ethical supervisory practices, and questionable research practices. We assessed whether their perceptions are related to career considerations. We gathered quantitative self-report estimations of the perceptions of PhD candidates using an online survey tool and then conducted descriptive and within-subject correlation analysis of the results. While most PhD candidates experience fair evaluation processes, openness, integrity, trust, and freedom in their research climate, many report lack of time and support, insufficient supervision, and witness questionable research practices. Results based on Spearman correlations indicate that those who experience a less healthy research environment (including experiences with unethical supervision, questionable practices, and barriers to responsible research), more often consider leaving academia and their current PhD position.

**Data Availability Statement:** All data files are available from the OSF database (DOI 10.17605/OSF.IO/BQX7V) https://osf.io/bqx7v/.

**Funding:** DL was funded by Vidi Grant 452-17-013 from the Dutch Research Council (NWO). The

## 1. Introduction

PhD candidates' attrition rates and time-to-degree-completion are, according to [1], important measurements of the efficiency and effectiveness of doctoral education. Higher attrition rates and longer times-to-degree go hand in hand [2], and lead to increased costs for PhD candidates and institutions [3] as well as a range of negative consequences for PhD candidates and their supervisors [4]. The average rate of attrition or prolongation of doctoral studies in Europe, Australia, and North America is estimated to be around 60% across multiple studies [5]. Precise numbers are difficult to obtain for doctoral programs in Europe [3], but according to a recent survey, an aggregate of 34% of doctoral candidates who enrolled in European institutions in 2009 did not graduate within six years [6]. The literature uses various terms for PhD attrition, such as "quit," "leave," "does not finish," or "drop-out." Throughout our paper, "quit" or "attrition" refer to not finishing one's PhD candidature, and "leave" refers to researchers leaving academia after gaining their PhD.

Many factors have been suggested as contributing to attrition and, more broadly, to intentions of leaving academia, including: funding, quality of supervision, scientific discipline, exposure to questionable research practices, institutional factors, organizational climate, involvement and socialization in the academic environment, community support, mental

funders had no role in study design, data collection and analysis, decision to publish, or preparation of the manuscript.

**Competing interests:** I have read the journal's policy and the authors of this manuscript have the following competing interests: All authors are employed by the university from which they gathered their data (i.e., the study is based on the responses of PhD candidates employed by Eindhoven University of Technology).

health, financial and nonfinancial costs, a lack of career prospects, and personal factors, such as life situations and attitudes towards doctoral studies [1–5, 7–17]. Most studies on PhD candidates who quit academia assume that this is undesirable and call for preventive measures; an assumption reflected by the value-laden term "attrition". Some literature questions this assumption, and emphasizes that there may be good reasons at early career stages to quit or leave academia, such as the (increasingly) uncertain longer-term prospects and (decreasing) availability of job resources [see e.g., 18, 19]. Although we acknowledge that low attrition is not necessarily desirable, our study focuses on the impact of factors connected to PhD students' research climate and their exposure to questionable research practices and questionable professional conduct, including poor supervision. Lowering attrition due to these factors is part of a university's duties of care.

Funding is often cited as one of the most robust predictors of doctoral completion [1, 20]. However, even in countries such as the Netherlands where most PhD candidates are considered employees and earn a salary, only a minority complete their PhD in the usually appointed four years and the average completion time is five years [15]. In the absence of funding-related concerns, van Rooij and colleagues [17] identified as a key determinant of candidates' intention to quit their PhD, the research climate. In their paper, research climate is related to such things as experienced workload, the quality of the (academic and personal) relationship with the supervisor, a sense of belonging, and the amount of freedom PhD candidates are granted to carry out their own project. Other studies also show the effects of supervisory behavior on PhD candidates' attrition rates [8, 11, 14, 16]. Experienced workload and supervision are sometimes directly related: in a Dutch nation-wide study [21] conducted by the national group representing PhD candidates' interests, Promovendi Netwerk Nederland (PNN), almost half (43%) of the PhDs reported that their supervisor engaged in one of nine supervisory behaviors that they labeled questionable based on previous reports of actions that put strain on PhDs. These included downplaying the workload or not recognizing work pressure (22%), contacting PhDs during weekends or evenings (17%), and pressuring PhDs to take on additional tasks (13%). The PNN study results are published in a range of reports, leading to several different citations within our paper. To unify our references to the same study, we refer to these reports here as part of the PNN study.

Factors relating to researchers' ethical and professional conduct are discussed less in the literature. Recent studies highlight that experience with questionable research practices may have a negative impact on the career of early career researchers [e.g., 22]. Questionable research practices (QRPs) are actions considered unethical by many (but not all) researchers, yet typically not considered as misconduct [23]. QRPs thus form a gray zone between good and bad practices and commonly involve some type of misrepresentation, inaccuracy, or bias [24], often specific to context, discipline, or subjective judgements. Existing taxonomies include a wide variety of QRPs [see e.g., 25–28]. Commonly used examples include not publishing negative results, rounding down *p* values, granting author status to non-contributing researchers ('gift authorship'), and insufficient supervision. In a recent study conducted with PhD candidates in the Netherlands and Sweden, Arlinghaus and Kekecs [7] found that 24% of their respondents considered leaving academia because of exposure to QRPs: PhD candidates' attrition considerations increased by a factor of 1.42 with each observed QRP.

The current study aims to find additional evidence for the importance of these different factors in explaining doctoral attrition. Additionally, given that the factors may be significantly correlated, we aim to contribute to the existing literature by examining these correlations and investigating the combined effects of: (1) the research climates in which PhD candidates operate; (2) their experiences of misconduct and questionable research practices; and (3) concerns related to supervisors' behavior. In the remainder of this paper, we first present the

quantitative questions we used in our online survey tool. After discussing our results in terms of descriptive statistics, we then shift our focus on how PhD candidates' experiences in their immediate research environment (i.e., the research climate where they work, questionable research practices, and supervisors' conduct) correlate with their considerations of leaving academia or quitting their PhD. Rather than estimating the true prevalence of subpar and questionable practices, we are interested in PhD candidates' subjective beliefs about the quality of their research environment. We conclude that PhD candidates who perceive more questionable practices and insufficient supervision feel their research environment is poor, and the poorer their perceptions about their research environment, the more often candidates consider leaving. Since our study concerns one Dutch university, it has both the drawbacks and the benefits of a case study; the generalizability of our results is limited, but the uniqueness of the Dutch doctoral education system and the relative homogeneity of the macro-level environment (nation- and university-wide policies) aid the detection of meso-level differences.

## 2. Context

The population studied are PhD candidates working at Eindhoven University of Technology (TU/e) in the Netherlands. They primarily engage in STEM (science, technology, engineering, and mathematics) research. STEM researchers constitute an understudied population [8] that might differ substantially from non-STEM populations, as natural and laboratory sciences appear to have (much) lower attrition rates than, for example, the social sciences and humanities [2, 4, 10].

PhD candidates in the Netherlands are generally employees and earn a salary, which means "funding" is much less of a concern for them. Studies show that PhD candidates who have more stable financial resources are more likely to complete their training [1–3, 10, 16]. In light of this, our study design allows us to focus on what we are most interested in, namely PhD candidates' experiences with their research environments. While we acknowledge limitations to generalizability (discussed in Section 5.1 below), our data provides an important starting point to examine correlations between factors that predict doctoral attrition among STEM PhD students.

According to the PNN study [29], despite most PhD candidates in the Netherlands having employee status, there are alternative legal arrangements such as scholarships or individually funded external positions. The typical duration of PhD contracts is 4 years. Contract hours range from 32 to 40 hours a week; scholarship and external PhD candidates are not usually bound to a formal number of working hours. The PNN study reported that PhD candidates' main obligations were research, taking courses, and teaching [30]. To complete their PhD, candidates are expected to produce a dissertation either in the form of a monograph or a collection of articles. Policies and exact guidelines on what is accepted for a dissertation differ depending on the university, research group or department, and discipline.

At the time of our study, the TU/e website listed 1,608 PhD candidates. As a comparison, in 2018, around 5,000 candidates completed their PhD in the Netherlands [31]. Given the university's disciplinary focus and existing support systems, along with the fact that most PhD candidates are funded, we did not expect to see high attrition rates.

## 3. Methods

### 3.1. Recruitment and participation

The planned methods, design, and analytical steps were registered in the study proposal and ethical form approved by the TU/e's Ethical Review Board (see the project's OSF repository for all supplementary materials as well as an explanation of all deviations from the steps discussed

in the original proposal: https://osf.io/bqx7v/). We hoped to recruit at least 200 participants based on feasibility considerations of the entire population, and the response rates of similar, previous studies; these rates were 3% in a university-wide Open Science Community Eindhoven and Information Expertise Center survey, and 21% in a nation-wide survey [32]. We wanted to describe the opinions of PhD students as the percentage of individuals who agree or disagree with statements, and estimate correlations between variables. After collecting 200 participants we would be able to describe frequencies with a margin of error of at most 7% (i.e., if 20% of participants would choose an answer option, the margin of error would range from 13.5% to 27.4%), and correlations with a margin of error of at most 0.14 (i.e., for a correlation of 0). Given the achieved sample size of 391, these maximum margin or errors decreased to 5% (for frequencies) and 0.1 (for correlations), which determine the granularity at which readers can interpret descriptive patterns in the data. For many descriptive statistics our interest is in how often people agree or disagree (regardless of whether they do so somewhat or strongly), or if questionable practices are observed at all, in which case accuracy in one response category is less important. Readers are advised to take the remaining uncertainty into account, especially when using our data for policy decisions.

All 1,608 PhD candidates listed as working at TU/e were invited and later reminded to participate via their official university email address (see OSF repository). To increase response rates and reduce non-response bias, email communication was designed based on best practice recommendations by Dillman and colleagues [33], and ten EUR 50 vouchers were raffled among the participants. Data collection took place from December 2020 to January 2021 via LimeSurvey (see OSF repository).

Duplicate responses, responses with data missing and only the demographic questions answered, were considered invalid and excluded from the data analysis. Sample size after removing invalid responses was 391 (32% women, less than 1% gender variant/non-conforming). The response rate was 24%, slightly higher than the 20% average in similar surveys [34]. Respondents had a mean age of 28.8 years, 95% confidence interval [28.1, 29.4]; 62% were 25 to 29 years old; 48% of the participants were Dutch, 33% indicated belonging to an ethnic minority, and 24% to a racial minority. The sample is similar to the TU/e doctoral population in terms of age (59% between 25 and 29 years old), nationality (41% Dutch), and gender (33% women). Similarity to racial and ethnic minority status could not be established due to lack of institutional data.

### 3.2. Survey design

After giving informed consent, participants were asked to complete five blocks of survey questions in the order as presented in the OSF repository. Unless stated otherwise, perceptions about frequency (ranging from 'never' to 'very often' for career considerations, and 'never' to 'almost always' for QRPs) and extent of agreement ('strongly disagree' to 'strongly agree') could be expressed on a 7-point Likert scale (for a complete list of questions, items, and scales, see OSF repository).

**Demographic and general data.** Participants were asked to report their age, gender, country of origin, nationality (Dutch/non-Dutch), the year they started their PhD and the year they expected to complete it, research area, and ethnic and racial minority status.

**Responsible research climate.** To measure to what extent participants experienced a responsible research climate, we included six facilitators and four barriers [35]. Many studies highlight the role of macro-level (e.g., structural problems within academia) and meso-level (e.g., organizational culture and climate) influences on unethical behaviors and research quality [36–44]. Haven and colleagues [35] distinguish "the shared meaning researchers attach to

the policies, practices and behaviors they associate with a responsible research climate" from general perceptions about the organization. By asking 61 researchers in focus-group interviews to reflect on the facilitators of and barriers to responsible research climates, the authors identified six facilitators (fair evaluation, openness, sufficient time, integrity, trust, and freedom) and four barriers (lack of support, unfair evaluation policies, normalization of overwork, and insufficient supervision). Haven and colleagues [35] use the more general term "characteristics," which we decided to change to "facilitators" to simplify our terminology and better indicate the opposing nature of barriers and characteristics of responsible research climates.

We considered these ten items the most relevant for our study for two reasons: First, they were derived from interviews conducted on a sample of researchers working in the Netherlands, which increases the likelihood that our participants would recognize the concepts. Second, the items are less connected to macro-level systemic issues and organizational structures that PhD candidates might have seldom or not directly experienced. To implement these characteristics, participants were asked to what extent they experienced the facilitator or barrier within their research environments.

**Questionable research practices.** Participants were asked to recall encounters and perceptions on the prevalence of misconduct and QRPs in their disciplinary fields, as well as in their current work environments (research group or institution). Scientific misconduct refers to any (attempted) action that undermines academic integrity, most typically fabrication, falsification, or plagiarism [23, 45]. To measure QRPs, we combined the ten highest-ranking items according to frequency and impact on trust, and the ten highest-ranking ones according to frequency and impact on validity from a sixty-item QRP taxonomy [25]. Removing duplicates resulted in a list of 14 items, such as "Not publishing a valid 'negative' study," "Ignoring basic principles of quality assurance," or "Not reporting clearly relevant details of study methods." To these, we added three other items: "Fabricate or falsify data," "Plagiarize" (according to the most common characterization of research misconduct), and "Other behaviors that I perceive as questionable research practices."

**Supervision.** To examine how doctoral attrition relates to supervisory practices, participants were asked about their supervisors' conduct. In terms of fostering research integrity, supervisors play a role as ethical examples [46], and are important for creating a responsible research climate [35]. According to professional guidelines, supervisors are expected to facilitate responsible research by providing competent supervision and mentoring [47]. Some supervisory actions are deemed unethical [48] or have detrimental consequences in terms of research integrity considerations [25, 35]. Based on a series of qualitative studies, Löfström and Pyhältö [46] listed five ethical principles underlying supervisory practices: respect of autonomy, non-maleficence, beneficence, justice, and fidelity. From these ethical principles, they constructed a 16-item Ethical Issues in Supervision Scale with subcategories such as exploitation, intrusion of supervisor views and values, inadequate supervision, inequality, and unfair authorship. Failure to adhere to these ethical principles was related to poor outcomes of the doctoral process in terms of engagement, satisfaction with supervision and doctoral candidacy, burnout, and attrition intentions.

The questions in our study were developed based on these Ethical Issues in Supervision Scales [46]. Participants first answered a set of questions on "How would you characterize your experience with your supervisor? If you have multiple supervisors, think about your primary supervisor, or the person you are most in contact with regarding your supervision. Please indicate the extent to which you agree with the following statements." Eighteen items such as "I receive supervision when I need it," and "My supervisor lacks cultural competency / multicultural sensitivity," were rated in terms of extent of agreement. Participants could agree or disagree to items attached to the second set of questions based on more serious ethical

violations listed by January and colleagues [49], and we asked "Do any of your supervisors engage in the following actions?" Six items such as "Have intimate relationships with a doctoral student," "Harassment (including unwanted sexual coercion)," and "Use slurs or make jokes about racial or ethnic groups" were presented. Items were rated "No," "Yes," or "Don't know / Prefer not to disclose."

**Career, attrition, and leave considerations.** Participants were asked about their career commitment and considerations of leaving academia or quitting their current job. To tap into different aspects of career considerations, we presented two sets of questions. First, using questions on career commitment developed by Blau [50], we asked participants to rate whether they agreed to seven statements such as "I definitely want a career for myself in academia" or "I am disappointed that I ever entered academia." Second, participants were asked: "How often have you seriously considered quitting academia / your current job?" based on questions previously presented by Spector and colleagues [51].

# 4. Results

We present descriptive results for all relevant survey questions in Figures and provide Tables with all relevant correlations between survey questions. When interpreting the data, we discuss general patterns and highlight the most important variables (either in terms of frequencies, or in terms of the size of the correlation). We did not test any hypotheses. Given the sample size of $N$ = 391 PhD students, correlations above 0.18, 0.21, 0.22, and 0.24 are statistically significant at the 0.05, 0.01, 0.005, and 0.001 level in a two-sided test, respectively. For the reader's convenience, we flag correlations significant at alpha levels of 0.05 and 0.001 in all correlation tables, so as to transparently communicate that there is a low probability we mistake random variation as a true effect.

## 4.1. Missing data

The overall percentage of missing data, including "don't know," "doesn't apply," and "prefer not to say" answers, was 15%. Missing data pertained especially to supervision-related questions about collaboration and favoritism, and to QRP for participants who recently became PhD candidates. This might reflect the lack of experiences with collaboration and QRPs among this sub-population (both in a disciplinary and work environment context). We provide a more in-depth analysis of missing data in our OSF repository.

## 4.2. Responsible research climate

Most participants had positive impressions about their research climate. Almost all agreed that they experienced fair evaluation, integrity, freedom, trust, and openness. Most agreed that they experienced sufficient time for work (Fig 1). Most seemed satisfied with the fairness of evaluation policies, support, and supervision, but almost half reported overwork being normalized within their research environment (Fig 2).

Both on the facilitator and barrier side, time constraints emerge as the most serious concern. One out of four respondents (25%) disagreed with having sufficient time for work; and almost 45% agreed with having experienced normalization of overwork within their scientific environment. Results of a Spearman correlation indicated a significant negative association between having sufficient time (facilitator) and normalization of overwork (barrier), (rS = -.47, $p$ < .001). One out of five respondents agreed with experiencing a lack of support (18%) and/or insufficient supervision (19%), and these two items were strongly positively correlated (rS = .69, $p$ < .001).

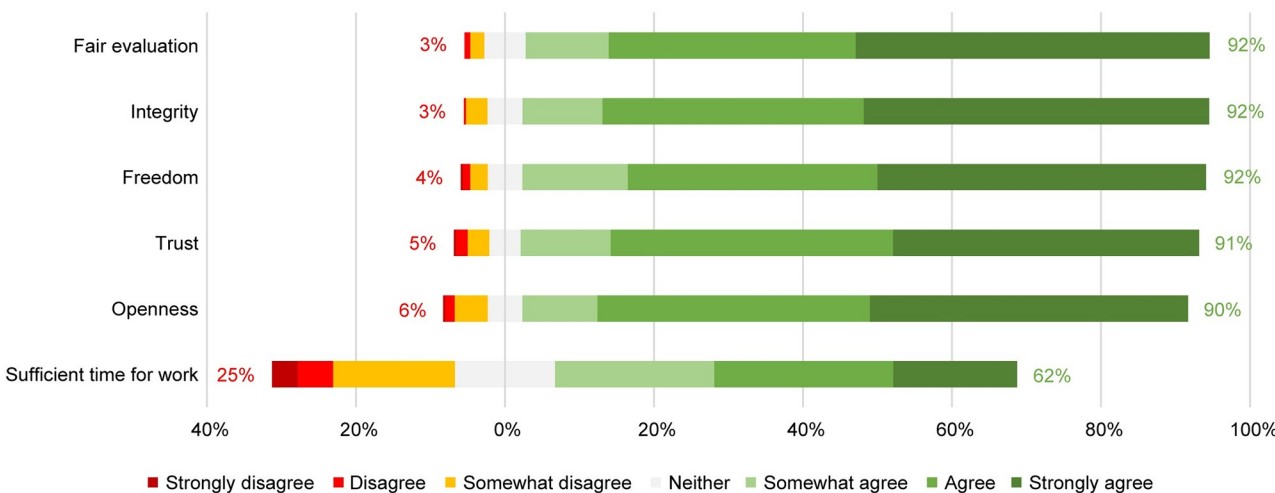

**Fig 1. Facilitators of a responsible research climate experienced in the scientific environment.**

In terms of correlations between items, high intercorrelations can be perceived within both the set of 6 facilitators ($\alpha = .81$) and the set of 4 barriers ($\alpha = .73$). Furthermore, facilitators and barriers were negatively correlated with each other: those experiencing barriers to a larger extent reported experiencing facilitators to a lesser extent (Table 1).

Research climate items were also correlated with most items measuring supervision and QRPs within the work environment. Perceptions about almost all QRPs within the work environment showed weak to moderate negative correlations with almost all facilitators, and weak to moderate positive correlations with almost all barriers (see OSF repository).

## 4.3. Questionable research practices

Respondents indicated that QRPs (Fig 3) were more prevalent within their disciplinary field than within their current work environment. The majority of PhD candidates ($N_{discipline} = 74\%$, $N_{work} = 63\%$) estimated that at least one QRP occurs *at least* very rarely within their discipline and their work environment. The three most frequently reported QRPs were stated by 54% of the participants (insufficiently reporting flaws or limitations, not reporting methods,

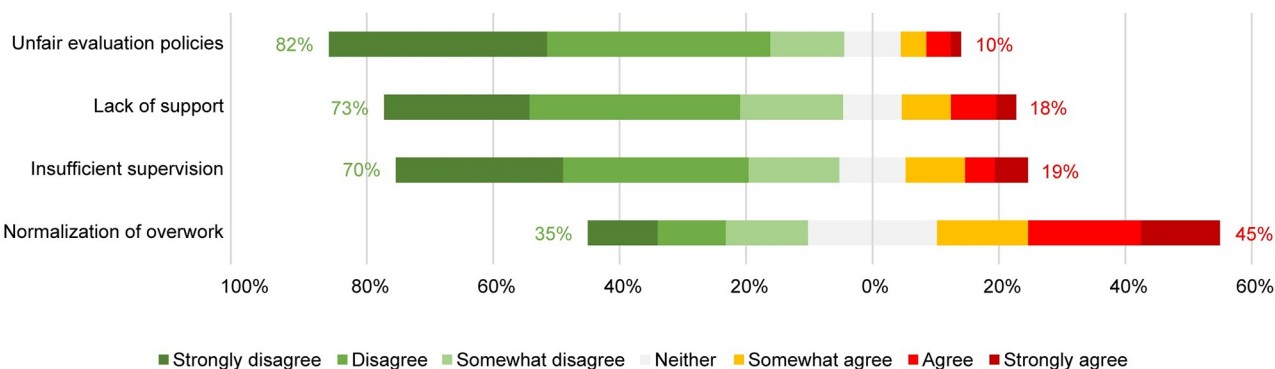

**Fig 2. Barriers to a responsible research climate experienced in the scientific environment.**

**Table 1. Correlations among research climate items.**

| Variables | Fair evaluation | Openness | Sufficient time for work | Integrity | Trust | Freedom | Lack of support | Unfair evaluation policies | Normalization of overwork |
|---|---|---|---|---|---|---|---|---|---|
| Openness | .62** | | | | | | | | |
| Sufficient time for work | .26** | .36** | | | | | | | |
| Integrity | .42** | .42** | .38** | | | | | | |
| Trust | .53** | .51** | .38** | .65** | | | | | |
| Freedom | .32** | .35** | .32** | .36** | .44** | | | | |
| Lack of support | -.45** | -.38** | -.32** | -.37** | -.44** | -.24** | | | |
| Unfair evaluation policies | -.54** | -.44** | -.24** | -.36** | -.44** | -.23** | ..54** | | |
| Normalization of overwork | -.19** | -.20** | -.47** | -.24** | -.19** | -.15* | .21** | .28** | |
| Insufficient supervision | -.46** | -.41** | -.27** | -.35** | -.38** | -.26** | .69** | .48** | .20** |

Note.

* p < .05

** p < .001 (2-tailed).

and selective citing, see Fig 3, panel A) for their discipline. The top three QRPs in their working environment reported by 39% and 38% of participants, were keeping inadequate notes, insufficient supervision, and insufficiently reporting flaws or limitations). In addition, publication bias (i.e., not publishing a valid 'negative' study) was the main practice reported occurring at least often, both within the discipline and the work environment.

About 20% of PhD candidates estimated that misconduct is at least a very rare occurrence within their discipline, while 2% estimated that misconduct occurs often, very often, or almost always within their discipline. About 10% reported occurrences of plagiarism or fabrication / falsification within their work environment.

## 4.4. Supervisory practices

Almost all respondents experienced receiving criticism in a friendly manner (90%), that they could tell supervisors if personal matters were affecting their work (86%), and that they could negotiate central choices for their dissertation (87%), see Fig 4. However, one out of five reported to have been left without supervision at one time (22%) and/or left without help if their supervisor could not advise them (22%).

A total of 32 respondents (8%) reported experiencing that their supervisors engaged in at least one serious transgression: racism (4%), sexism (3%), slurs/jokes about other minorities (3%), intimate relationships with PhD candidates (2%), harassment (less than 1%), and homophobia (less than 1%). Respondents reported a total of 53 such serious transgressions, with 19 PhD candidates reporting only one. Note that it is possible multiple PhD students independently reported observing the same transgression.

## 4.5. Career considerations

The majority of PhD candidates did not express strong negative feelings about academia as a vocation. Responses to questions about future career paths were more mixed. For example,

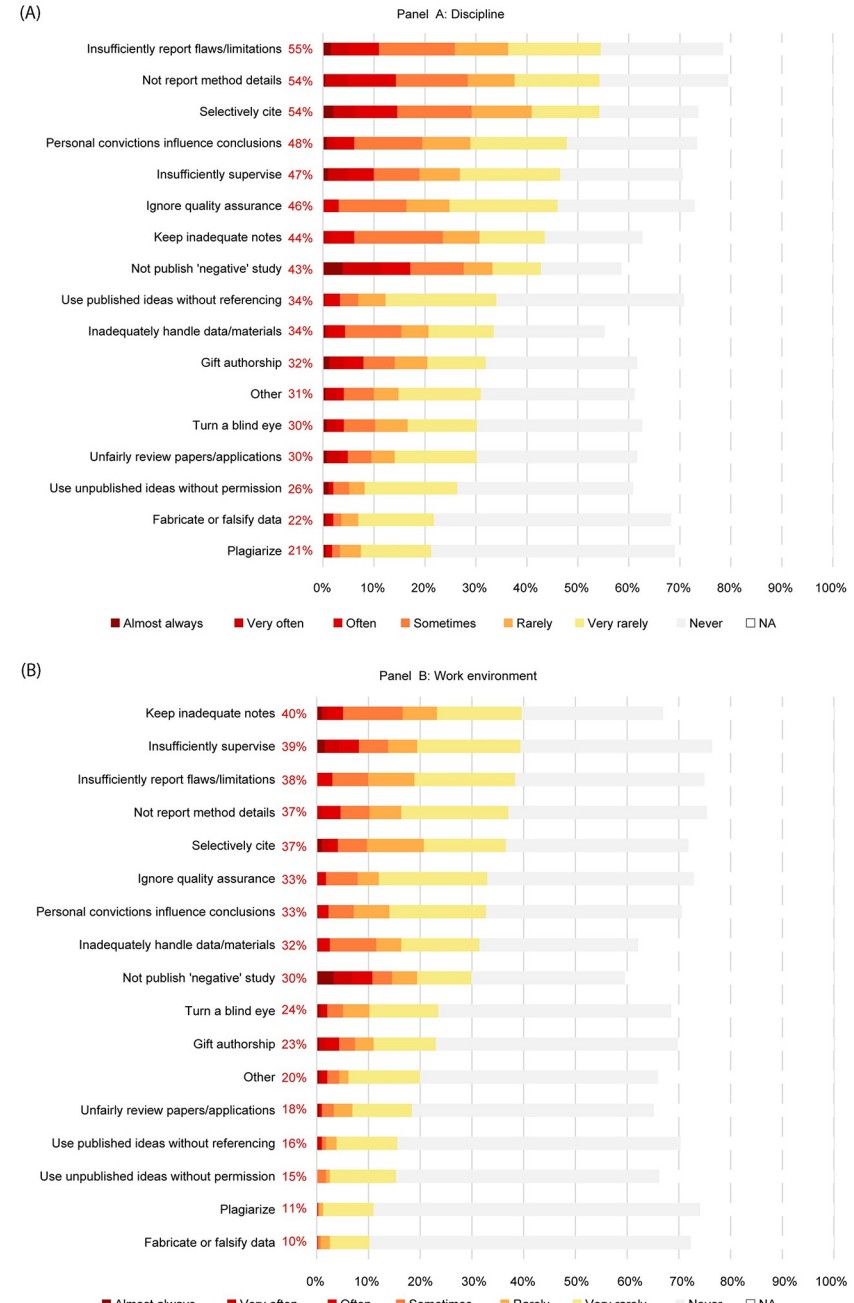

**Fig 3. Estimated Prevalence of Questionable Research Practices, in the discipline (panel A) and the work environment (panel B).** *Note*: Item descriptions were shortened for this figure. Full descriptions are in Table 8. Numbers in red are rate of respondents reporting occurrences "very rarely".

50% reported they loved doing research in academia too much to give it up and 8% were disappointed that they had entered academia (Fig 5).

Respectively, 10% and 18% of PhD candidates considered quitting their current job and leaving academia at least often (Fig 6). Not surprisingly, respondents who started working at the university more recently, had less frequently considered leaving their job ($r_S$ = -.25, p <

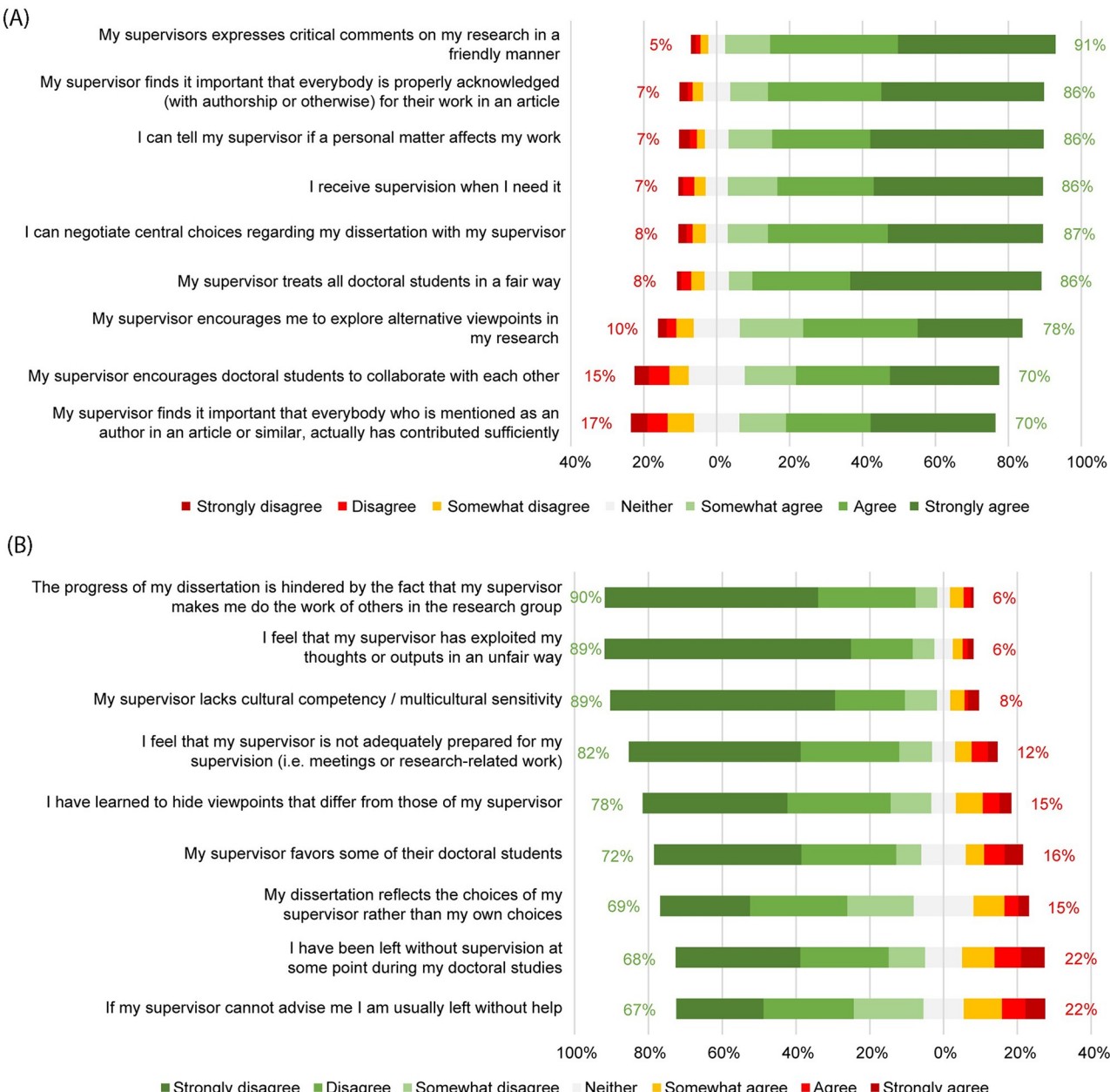

**Fig 4. PhD candidates' characterization of their primary supervisor's practices.**

.001) as well as academia ($r_S$ = -.34, p < .001). Results of a Spearman correlation indicated a significant positive correlation between considerations of quitting the current job and of leaving academia ($r_S$ = .57, p < .001). This correlation is high, and significant, but it is not perfect: in some cases, PhD students wanted to leave academia but not their current job as PhD students.

## 4.6. Demographic subgroup analyses

Experiences with research climates did not differ significantly based on ethnic or racial minority status and research area. There were some gender differences: female respondents (50%)

(A)

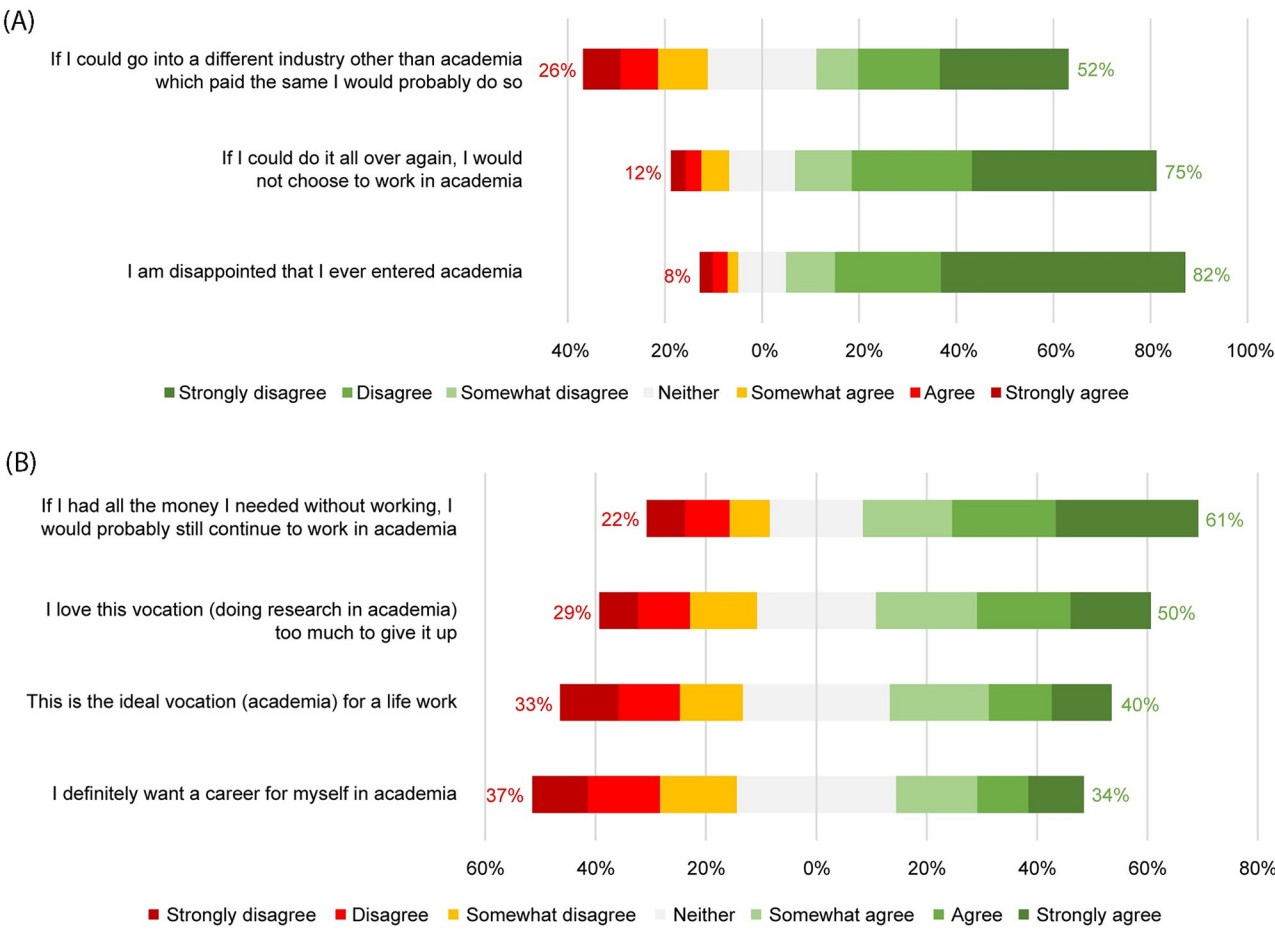

**Fig 5. Vocational commitment.**

were more likely to report normalization of overwork than male respondents (35%). Most of the statistically significant differences were related to nationality (Fig 7).

Similar to experiences with research climates, perceptions about supervisory practices varied between Dutch and non-Dutch PhD candidates (Fig 8).

Non-Dutch PhD candidates were more likely than Dutch PhD candidates to report: definitely wanting a career in academia (40% vs 24%), love doing research in academia too much to give it up (45% vs 30%), and feeling academia was the ideal vocation (11% vs 3%).

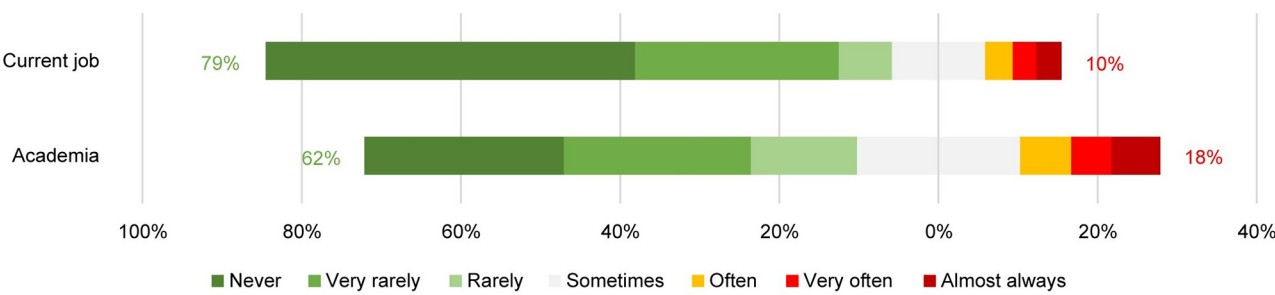

**Fig 6. Frequency of PhD candidates seriously considering quitting their current job and academia.**

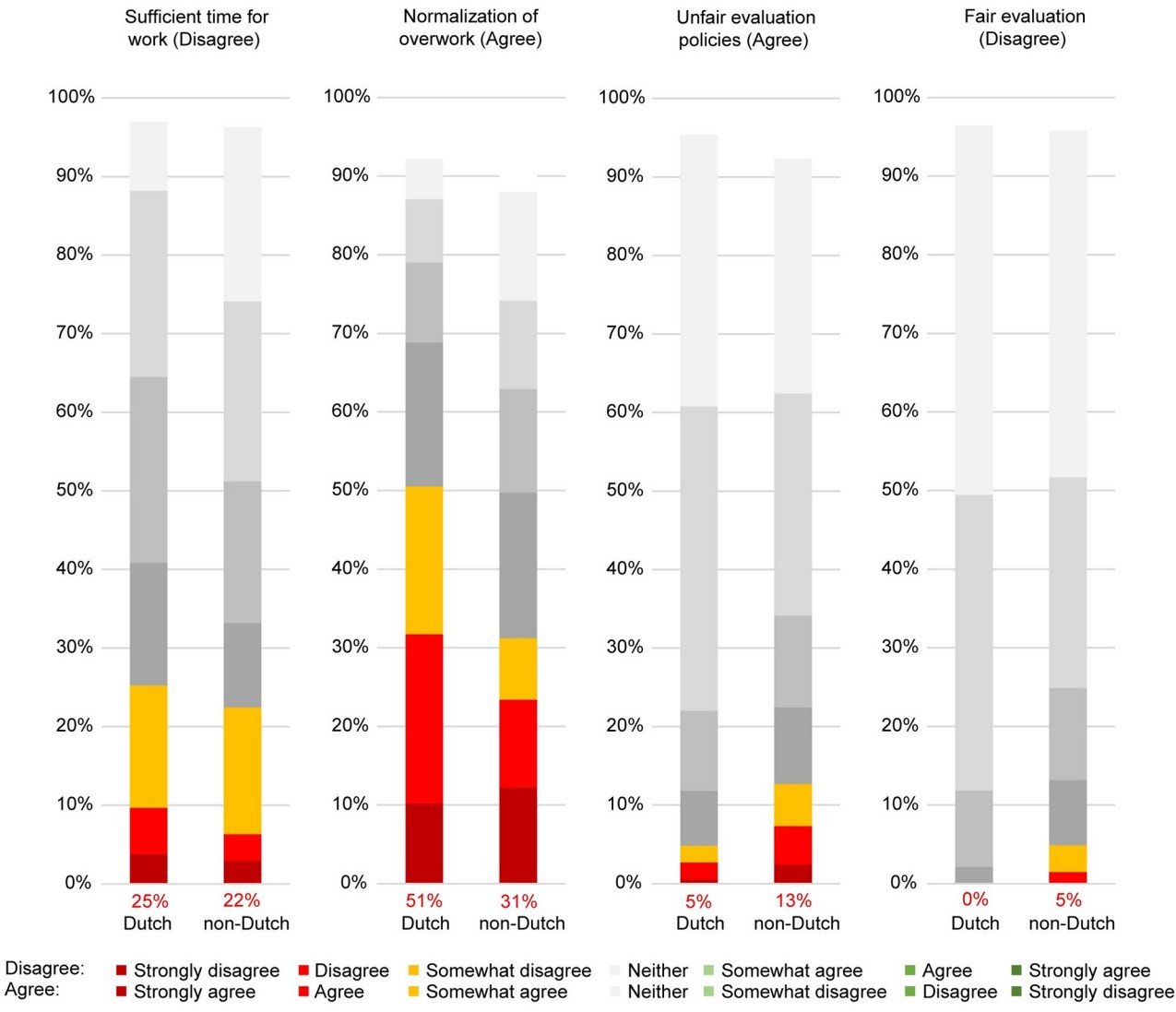

**Fig 7. Rate of Dutch and non-Dutch PhD candidates who report experiencing sufficient time for work, unfair evaluation policies, or normalization of overwork.** *Note*: Scales range either from strongly disagree to strongly agree (marked "disagree") or the opposite, strongly agree to strongly disagree (marked "agree"). Numbers in red represent the rate of respondents who at least "somewhat agree/disagree".

## 4.7. Correlations between research climates, QRPs, and supervisory practices

Perceptions about nearly all QRPs within the work environment correlated negatively with almost all facilitators, and positively with almost all barriers of responsible research climates (i.e., correlations were significant between almost all items). All significant correlations were weak (<0.4) to moderate (0.4 to 0.6) (Table 2) according to the naming convention most commonly used in psychology [52]. Integrity and trust (research-climate characteristics) were related to all questionable practice items. This substantiates the idea that, although QRPs might form an ethical gray zone, they are perceived to be connected to a lack of research integrity. In addition, all research climate items were significantly correlated to the only supervision-related questionable practice (i.e., insufficient supervision). Most supervision-practice items were also related to perceptions about research climates (Tables 3 and 4).

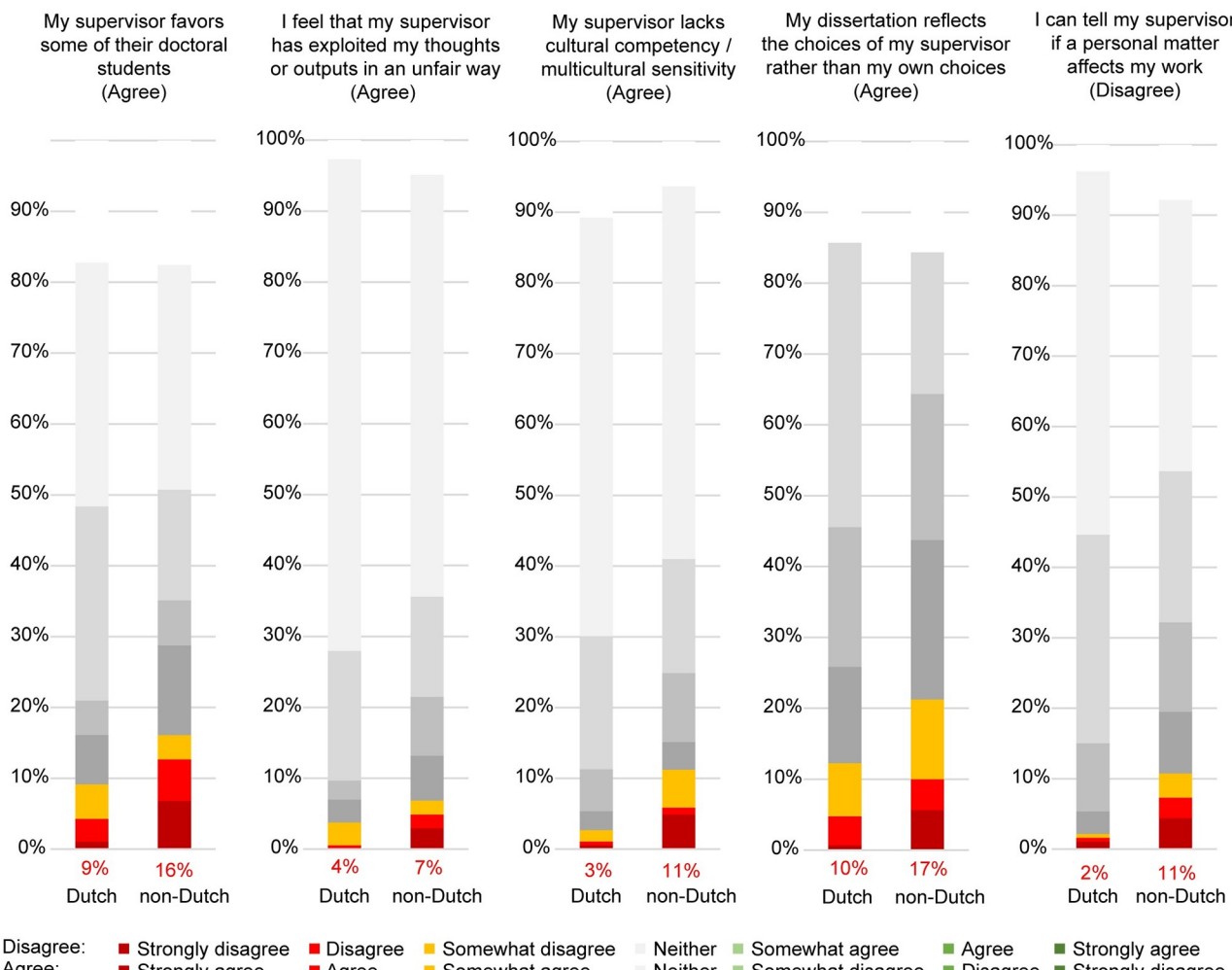

**Fig 8. Dutch and non-Dutch PhD candidates' characterization of their primary supervisor's practices.** *Note*: Scales range either from strongly disagree to strongly agree (marked "disagree") or the opposite, strongly agree to strongly disagree (marked "agree"). Numbers in red represent the rate of respondents who at least "somewhat agree/disagree".

## 4.8. Correlations with leave and quit considerations

In terms of responsible research climates, both leave and quit considerations correlated negatively with facilitators (e.g., trust or freedom), and positively with barriers (lack of support or insufficient supervision), see Table 5. Most supervisory practices were also related to leave and quit considerations. For example, experiences with actions like being left without supervision or supervisors' inadequate preparedness were positively correlated with leave considerations (Table 6), while fair treatment of all PhD candidates and the ability to negotiate central choices for the dissertation were negatively correlated with leave and quit considerations (Table 7). These correlations, even when significant, are weak. This indicates that, even though the correlations are related, other factors might influence the decision to quit or leave, or possibly a combination of factors. This analysis, however, falls outside the scope of this paper, and could be addressed in future research.

**Table 2. Spearman correlations for research climate items and questionable research practices within the work environment.**

| | Inadequately handle / store data | Keep inadequate notes | Ignore basic quality assurance | Not publish a valid 'negative' study | Not report relevant details of methods | Selectively cite | Let personal convictions influence conclusions | Insufficiently report study flaws and limitations | Turn a blind eye | Use unpublished ideas / phrases without permission | Use published ideas / phrases without referencing | Unfairly review papers, grant applications | Insufficient supervision | Gift authorship | Fabricate or falsify data | Plagiarism | Other |
|---|---|---|---|---|---|---|---|---|---|---|---|---|---|---|---|---|---|
| Fair evaluation | -.27** | -.33** | -.39** | -.26** | -.30** | -.24** | -.25** | -.25** | -.33** | -.23** | -.19* | -.31** | -.42** | -.25** | -.14* | -.15* | -.27** |
| Openness | -.25** | -.32** | -.38** | -.29** | -.31** | -.26** | -.27** | -.29** | -.28** | -.23** | -.22** | -.29** | -.35** | -.25** | -.16* | -.20** | -.24** |
| Sufficient time for work | -.22** | -.25** | -.25** | -.36** | -.28** | -.24** | -.23** | -.23** | -.15* | -.08 | -.06 | -.23** | -.26** | -.19* | -.03 | -.04 | -.20* |
| Integrity | -.38** | -.37** | -.45** | -.35** | -.44** | -.35** | -.38** | -.38** | -.39** | -.27** | -.25** | -.28** | -.38** | -.34** | -.24** | -.24** | -.36** |
| Trust | -.33** | -.37** | -.42** | -.32** | -.38** | -.30** | -.29** | -.34** | -.34** | -.24** | -.20** | -.34** | -.38** | -.33** | -.20** | -.23** | -.30** |
| Freedom | -.20* | -.21** | -.17* | -.15* | -.17* | -.20** | -.17* | -.15* | -.16* | -.07 | -.07 | -.10 | -.14* | -.19* | -.02 | -.03 | -.13* |
| Lack of support | .25** | .32** | .32** | .24** | .30** | .22** | .27** | .28** | .23** | .18* | .18* | .22** | .40** | .23** | .12* | .13* | .29** |
| Unfair evaluation policies | .27** | .26** | .38** | .29** | .31** | .24** | .25** | .23** | .28** | .24** | .23** | .28** | .34** | .26** | .21** | .20** | .28** |
| Normalization of overwork | .22** | .23** | .23** | .22** | .20** | .16* | .24** | .20** | .14* | .04 | .12 | .14* | .23** | .16* | .07 | .03 | .14* |
| Insufficient supervision | .27** | .33** | .37** | .26** | .28** | .20** | .23** | .28** | .20* | .17* | .19* | .19* | .42** | .19* | .12* | .12* | .26** |

**Table 3. Spearman correlations for research climate items and supervisory practices (A).**

| | Left without supervision at some point | If supervisor cannot advise, left without help | Favors some of their PhDs | Exploited thoughts / outputs | Progress hindered because made to do others' work | Learned to hide differing viewpoints | Dissertation reflects choices of supervisor | Inadequately prepared for supervision | Lacks cultural competency / multicultural sensitivity |
|---|---|---|---|---|---|---|---|---|---|
| Fair evaluation | -.37** | -.32** | -.32** | -.38** | -.29** | -.37** | -.28** | -.37** | -.41** |
| Openness | -.29** | -.27** | -.25** | -.37** | -.27** | -.28** | -.19** | -.35** | -.33** |
| Sufficient time for work | -.18** | -.16* | -.08 | -.10 | -.17** | -.08 | -.11* | -.23** | -.10 |
| Integrity | -.27** | -.28** | -.30** | -.28** | -.27** | -.24** | -.18** | -.32** | -.32** |
| Trust | -.28** | -.23** | -.27** | -.33** | -.29** | -.33** | -.23** | -.31** | -.29** |
| Freedom | -.19** | -.19** | -.21** | -.27** | -.21** | -.29** | -.37** | -.24** | -.27** |
| Lack of support | .47** | .42** | .33** | .32** | .30** | .35** | .28** | .46** | .27** |
| Unfair evaluation policies | .29** | .25** | .38** | .44** | .34** | .35** | .27** | .36** | .38** |
| Normalization of overwork | .11* | .11* | .07 | .07 | .21** | .07 | .14* | .15* | .14* |
| Insufficient supervision | .62** | .54** | .37** | .32** | .28** | .36** | .29** | .52** | .33** |

Most QRPs in the discipline were not significantly correlated with leave or quit considerations, while most QRPs within the work environment were weakly positively correlated (Table 8).

## 5. Discussion and conclusion

Training PhD candidates is an investment that requires funding, specific educational programs organized by the university, as well as supervisors' time and commitment. Negative

**Table 4. Spearman correlations for research climate items and supervisory practices (B).**

| | Receive supervision when need it | Can negotiate central choices for dissertation | Encourages PhDs to collaborate with each other | Encourages me to explore alternative viewpoints | Treats all PhDs fairly | Expresses critical comments in friendly manner | Only those attributed who contributed | Those who contributed are attributed | Can tell supervisor if personal matters affect work |
|---|---|---|---|---|---|---|---|---|---|
| Fair evaluation | .45** | .45** | .34** | .34** | .51** | .41** | .31** | .38** | .35** |
| Openness | .44** | .39** | .37** | .38** | .41** | .41** | .32** | .39** | .37** |
| Sufficient time for work | .24** | .14* | .38** | .25** | .18** | .10 | .25** | .20** | .19** |
| Integrity | .33** | .31** | .27** | .30** | .44** | .32** | .34** | .38** | .23** |
| Trust | .37** | .39** | .28** | .32** | .45** | .30** | .31** | .36** | .35** |
| Freedom | .27** | .30** | .29** | .35** | .27** | .35** | .20** | .22** | .29** |
| Lack of support | -.48** | -.36** | -.27** | -.30** | -.44** | -.24** | -.26** | -.23** | -.25** |
| Unfair evaluation policies | -.28** | -.37** | -.20** | -.29** | -.47** | -.30** | -.28** | -.30** | -.27** |
| Normalization of overwork | -.08 | -.10 | -.25** | -.13* | -.09 | -.15* | -.18* | -.10 | -.05 |
| Insufficient supervision | -.60** | -.43** | -.30** | -.34** | -.44** | -.26** | -.33** | -.30** | -.21** |

**Table 5. Spearman correlations for research climate items and leave/quit considerations.**

|  | Fair evaluation | Openness | Sufficient time for work | Integrity | Trust | Freedom | Lack of support | Unfair evaluation policies | Normalization of overwork | Insufficient supervision |
|---|---|---|---|---|---|---|---|---|---|---|
| Leave academia | -.26** | -.24** | -.29** | -.22** | -.27** | -.25** | .31** | .18** | .17* | .25** |
| Quit current job | -.18** | -.19** | -.17** | -.25** | -.20** | -.200** | .28** | .19** | .13* | .31** |

Note:

$^*$ p < .05,

$^{**}$ p < .001 (2-tailed).

impressions about research practices in the disciplinary field and the immediate work environment can affect science and academia's reputation, as well as research institutes'. Moreover, researchers have a professional responsibility towards PhD candidates to facilitate adequate supervision and contribute to a healthy work environment.

Our results corroborate several findings in the literature concerning factors for doctoral attrition, pertaining to experiences of: the research climates where PhD candidates operate (Section 4.2); misconduct and questionable research practices (Section 4.3); and supervisory practices (Section 4.4).

To these, we added insights in the *correlations* between such factors (Tables 2 to 4) and their correlations with doctoral attrition (Sections 4.7 and 4.8; Tables 5 to 8). In all the factors studied, we found some correlations of moderate strength (here defined as Spearman correlation > .4); correlations of factors with doctoral attrition were weaker (although with several items >.3). This strengthens intuitions on the combined effects of factors studied more qualitatively [35], and with small samples [7]. Overall, many of our constructs are moderately correlated. Although such correlations should be interpreted with caution, establishing them by combining constructs linked to professional and ethical conduct is a substantive contribution to the literature on doctoral attrition.

In terms of facilitators and barriers for responsible *research climates*, the majority of PhD candidates report experiencing fair evaluation, integrity, freedom, trust, and openness. Insufficient supervision and having insufficient time are frequently reported barriers to responsible research climates, barriers for which we found positive correlations with leave and quit

**Table 6. Spearman correlations for supervision items and leave/quit considerations (positive correlation).**

|  | Left without supervision at some point | If supervisor cannot advise, left without help | Favors some of their PhDs | Exploited thoughts / outputs | Progress hindered because made to do others' work | Learned to hide differing viewpoints | Dissertation reflects choices of supervisor | Inadequately prepared for supervision | Lacks cultural competency / multicultural sensitivity |
|---|---|---|---|---|---|---|---|---|---|
| Leave academia | .28** | .24** | .13* | .14* | .01 | .27** | .17* | .24** | .20** |
| Quit current job | .29** | .29** | .21** | .14* | .11* | .28** | .21** | .34** | .26** |

Note:

$^*$ p < .05

$^{**}$ p < .001 (2-tailed).

**Table 7. Spearman correlations for supervision items and leave/quit considerations (negative correlation).**

| | Receive supervision when need it | Can negotiate central choices for dissertation | Encourages PhDs to collaborate with each other | Encourages me to explore alternative viewpoints | Treats all PhDs fairly | Expresses critical comments in friendly manner | Only those attributed who contributed | Those who contributed are attributed | Can tell supervisor if personal matters affect work |
|---|---|---|---|---|---|---|---|---|---|
| Leave academia | -.26** | -.21** | -.24** | -.19** | -.26** | -.18** | -.12* | -.13* | -.29** |
| Quit current job | -.30** | -.31** | -.26** | -.24** | -.26** | -.23** | -.20** | -.21** | -.26** |

Note:

* p < .05

** p < .001 (2-tailed).

**Table 8. Correlations between perceived prevalence of questionable research practices within the discipline and work environment, and considerations about leaving academia or quitting.**

| Questionable research practices | Discipline | | Work env. | |
|---|---|---|---|---|
| Leaving | Academia | Job | Academia | Job |
| Inadequately handle or store data or (bio)materials—including inadequately archiving for an appropriate period of time | .12 | .07 | .15* | .15* |
| Keep inadequate notes of the research process—not keeping notes in (digital) lab journals or their equivalent in other types of research | .14* | .09 | .22** | .26** |
| Ignore basic principles of quality assurance | .12* | .12* | .24** | .27** |
| Not publish a valid 'negative' study (e.g., a well-designed study that did not confirm a solid theoretical prediction)—either in a journal, or a as a publicly available document | .16* | .17* | .22** | .24** |
| Not report clearly relevant details of study methods | .09 | .10 | .16* | .20** |
| Selectively cite to enhance ones' own findings or convictions | .12 | .15* | .15* | .19* |
| Let personal convictions influence the conclusions substantially | .10 | .05 | .16* | .15* |
| Insufficiently report study flaws and limitations | .11 | .01 | .15* | .16* |
| Turn a blind eye to supposed breaches of research integrity by others | .04 | -.00 | .13* | .06 |
| Use unpublished ideas or phrases of others without their permission—e.g., from reviewing manuscripts or grant applications, or from conference presentations | .01 | -.06 | .00 | .04 |
| Use published ideas or phrases of others without referencing | .03 | -.07 | .12 | .06 |
| Unfairly review papers, grant applications or colleagues' application for promotion | .17* | .05 | .09 | .03 |
| Insufficiently supervise or mentor junior coworkers | .27** | .17* | .25** | .26** |
| Demand or accept an authorship for which you don't qualify ['honorary or gift authorship'] | .10 | .02 | .11 | .14* |
| Fabricate or falsify data | -.04 | -.07 | .04 | .03 |
| Plagiarize | -.01 | -.04 | .03 | .05 |
| Other behaviors that I perceive as questionable research practices | .13* | .03 | .14* | .11 |

Note:

* p < .05

** p < .001 (2-tailed).

Considerations about leaving academia (leave) or current job (quit).

considerations. These results underline findings from previous research, that time constraints and supervisory practices are important influencers of doctoral satisfaction and attrition rates in the Netherlands [17, 53]. Lack of time is often cited as being detrimental to research integrity, the quality of research, education, mentorship, and a healthy work-life balance [37, 54]. We found intercorrelations within both the set of six facilitators ($\alpha$ = .81) and the set of four barriers ($\alpha$ = .73), indicating that they tend to co-occur. We also found significant correlations between perceptions of research climate, supervisory practices, and QRPs. This correlative relationship with variables related to responsible practices confirms the validity of the responsible research climates' construct by Haven and colleagues [35].

Respondents estimate that *questionable research practices* are more prevalent within their disciplinary field than within their current work environment. This is in line with earlier findings, where researchers report more wrongdoings when asked about (distant) others than themselves or their close environments as described in Fanelli's [55] meta-analysis. Still, a significant ratio of PhD candidates report experiencing questionable practices and even misconduct within their immediate surroundings. These outcomes are not directly related to actual prevalence estimates, as the reports were based on subjective perceptions and may not reflect actual misconduct or questionable research practices. Moreover, given their shared environment, several respondents might refer to the same perceived incident. Yet the reported incident rates warrant further investigation of the occurrence and effects of QRPs in more detail, since their impact could be large.

Based on our correlational outcomes, experiences with QRPs are positively related to experiences with barriers, and negatively to experiences with facilitators of responsible research climates. This confirms the idea that good things tend to co-occur (and bad things too): worse research climates also tend to have worse research practices. Indeed, trust and integrity are facilitators of responsible climates, but also fundamental for responsible research practices.

Leave and quit considerations were connected to most QRPs within work environments but not to those within the PhD candidates' disciplines. This may be an effect of distance: candidates are likely to be more upset by negative experiences closer to home than by factors experienced more indirectly. While our results are only correlative, they support the notion that experiences of QRPs are connected to considerations of leaving academia [7].

For the correlations between supervisory practices and quit and leave considerations, we found that the PhD candidates who have more extensive experiences with unethical supervision practices reported considering leaving and quitting academia with a higher frequency. This corroborates earlier findings that claimed a connection between attrition and supervision-related experiences [2, 5, 46, 53]. In addition, experiences with supervision display the strongest connections with perceptions about research climates—a result that is in line with the contention that insufficient supervision is an important barrier to responsible research climates [35].

Our results were obtained for a relatively homogeneous sample population, i.e., PhD candidates at a Dutch technical university. This partly addresses the lack of studies specific to STEM researchers [8], and suggests some insights related to confounding factors. Earlier work suggests that attrition rates in natural sciences are generally lower than in the social sciences and humanities [2, 4, 10]. In comparison, our study partly confirms this expectation: attrition considerations are indeed less prevalent in our sample (21%) than in the PNN study (42%; 53). However, first- and second-year candidates (overrepresented in our sample) may also have had less time to experience reasons for leaving (relating to research climate, QRPs, and supervision) than the candidates in the PNN study, who were more equally distributed over every stage of a PhD candidacy.

Moreover, 34% of the respondents in our survey indicated that they definitely want a career in academia, whereas the national survey reports a substantially higher percentage, 61.1. Some of this could be due to different formulations of the question (in our study: "I definitely want a career for myself in academia", rather than "In what sector would you like to work after obtaining your PhD?"). Still, the magnitude of the difference suggests that other factors might be at work. One explanation could be our candidates' stronger link to industry: STEM doctorate holders report the facilitating role of the connections they develop even during their PhD work [56], and the higher salaries and better working conditions offered by industry [57].

## 5.1. Limitations

Our study involved mainly first- or second year PhD candidates working at a technical university in the Netherlands. This restricts the generalizability of our results. PhD candidates' perceptions and concerns may differ at various stages of their career; and their experiences—especially in terms of exposure to supervisory and research practices—can be expected to change and diversify over time. Leave and quit considerations could be affected by the university's focus, since attrition rates are lower for students working in natural and laboratory sciences than for students in the humanities and social sciences [10]. As for national characteristics: most PhD candidates in the Netherlands are salaried employees. Hence it is currently unclear whether our descriptive findings apply generally to other populations, especially in other countries or less STEM-focused environments. Still, our findings were in line with previous observations in different populations, and we expect that the correlations observed in our sample might hold in other populations, but perhaps with a varying effect in size.

We relied on volunteer self-reports. We tried to implement motivational tools within our invitation procedure to increase the response rates and diversify the incentives underlying participation. We increased the benefits of participation by providing financial incentives, asking peers for help and advice, and conveying that other PhD candidates had already helped on prior occasions (see OSF repository). Still, self-selection might have led to overrepresentation of some respondent characteristics. For example, PhD candidates who are dissatisfied could be more likely to respond; or many respondents may be motivated to participate in similar studies because of a general interest in topics related to research integrity.

The measurements we used also impose limitations. Unlike previous studies, we asked participants about their career considerations rather than their intentions. The two concepts seem intertwined but are somewhat different, thus making comparisons to studies using intentions or behaviors as attrition measurements difficult. In addition, as our focus was on subjective perceptions (for which considerations are relevant), comparisons between attrition rates in other studies and our data are problematic. Similarly, our results do not pertain to objective prevalence estimates of questionable research practices or supervisory actions. Even if our respondents' reports could be considered objective estimates, one bad action could have been perceived and reported by multiple respondents. In addition, although we carefully selected the QRP items included in our study, we did not ask participants about their understanding of each item, nor their estimates of severity. These constraints affect both the generalizability of our results and the comparability with similar studies.

The correlations observed in our dataset do not allow for claims about causality. On the one hand, the fact that barriers and facilitators of research climates had strong inter-correlations with questionable (supervisory) practices and attrition or leave considerations could mean that bad experiences are more likely to occur in poor research climates. The opposite is also plausible, in that bad research climates lead to a deterioration of individual behaviors. From this

perspective, better practices could improve research climates, or research climates could facilitate better practices. On the other hand, these correlations could also be attributed to common method biases. A range of potential causes such as social desirability, common rater effects, or participants' biased mood could strengthen or weaken the correlations between measurements [58]. For example, good impressions of a supervisor might make participants blind to the transgressions the supervisor performs. Bad experiences might in turn curtail all the positive characteristics of the immediate environment. We recommend that future researchers examine the causality and then to what extent these experiences influence attrition considerations.

## 5.2. Recommendations

An increasing number of measures is being implemented in many universities, including TU/e, to improve the wellbeing of PhD candidates and reduce doctoral attrition. Many implications of our results are in line with recommendations in other studies and with policies that already have been partly implemented. These include: enhancing knowledge and skills related to the management of the doctoral candidacy and interactions within professional contexts [17]; ensuring a good fit between PhD candidates and their supervisors, both personally and professionally [59]; and providing sufficient guidance while also leaving room for PhD candidates to pursue their own research interests [60].

Several other implications emerged in conversations with university stakeholders about our results. Here, we summarize some of the recommendations we formulated during this process. Our study gives an indication of the link between PhD candidates' worst research integrity-related experiences and their considerations of leaving academia or quitting their current job. While connections between what PhD candidates experience in terms of integrity, ethicality, professional norms, and their career considerations have been made before, studies (and especially large sample quantitative studies) with an explicit focus on the matter at hand are scarce and fragmented. To gain further insights, we recommend that universities gather longitudinal data by conducting periodical surveys similar to the one presented in our paper. Future research using exit interviews should measure how often PhD candidates quit because of their experiences with unethicality, breaches of research integrity, or a general lack of responsible research. In addition, establishing the prevalence of wrongdoings and the number of connected wrongdoers would help to differentiate the impact of one outstanding offender and widespread issues. Further understanding of what and why PhD candidates perceive as a violation of professional codes of conduct or ethicality could also advance our understanding of how doctoral candidates are impacted by their experiences. Finally, we suggest periodical inquiries about PhD candidates' immediate research environments. If conducted and interpreted by university officials, such longitudinal data might provide valuable opportunities for interventions and insights into scarcely researched questions such as how and when attrition considerations develop.

Differences between the experiences of Dutch and non-Dutch PhD candidates can point to differences in the understanding of norms and expectations. Dutch candidates are more likely to feel the time pressures attached to employment, while not communicating norms explicitly could lead to inequalities in evaluation and expectations. Clarification of evaluation policies has a similar function and is particularly important for non-Dutch PhD candidates who already have to cope with cultural and language barriers. While we did not collect data on the characteristics of supervisors with whom PhD candidates had experiences relating to a lack of cultural sensitivity, both Dutch and non-Dutch supervisors and PhD candidates could benefit from a stronger expectation of building intercultural competencies. Here we also note that time pressure and workload are persistent and major focal points for all PhD candidates

regardless of nationality. In this regard, we reiterate the recommendations by van Rooij and colleagues [17]: an important step in reducing workloads would be to emphasize that it is not normal or expected to work outside contract hours (weekends, holidays, or evenings) and then take further precautions and additional steps to ease pressure.

Our results establish an estimate of the subjective prevalence of certain unethical supervisory practices and corroborate notions of the importance of good supervision. For this reason, we agree with the Netherlands PhD network recommendations advocating a commitment on the side of research institutions to prevent unethical supervisory behavior by providing training, setting norms and expectations, and implementing some type of evaluation system for supervisors [21]. With clear expectations, supervisors might have to take up new responsibilities or adjust their modus operandi in compliance with guidelines or according to PhD candidates' needs. These efforts should not go unnoticed. Universities could provide proper recognition of both daily and other supervisors and facilitate the spread of good practices by setting up peer discussion groups or semi-formal interactive forums for discussing supervisory practices.

Although research integrity training is available and mandatory for all doctoral candidates, no comparable training is required for supervisors. Even if training sheds light on the complexity of ethical decisions, PhD candidates have to (or are at least strongly incentivized to) work with the methods made available or accepted by their supervisors. Young, relatively inexperienced researchers might also have difficulty speaking up against, or even recognizing questionable research practices. Outcomes of recent studies indicate that an alarming percentage of PhD candidates report intentions to commit fraud driven by supervisors' norms [61]. Research integrity courses for or together with supervisors could help to open up discussions about disciplinary norms and set a basic level of understanding about (in)appropriate practices. We agree with the suggestion by Bouter and colleagues (2016): supervisors and PhD candidates might greatly benefit from a list of questionable research practices for training and supervisory purposes.

## Supporting information

**S1 File.**
(DOCX)

## Acknowledgments

We thank József Lázár for assistance with gathering the list of email addresses of potential participants.

## Author Contributions

**Conceptualization:** Andrea Kis, Daniël Lakens.

**Data curation:** Andrea Kis.

**Formal analysis:** Andrea Kis.

**Funding acquisition:** Elena Mas Tur, Daniël Lakens, Krist Vaesen, Wybo Houkes.

**Investigation:** Andrea Kis.

**Methodology:** Andrea Kis, Elena Mas Tur, Daniël Lakens, Krist Vaesen, Wybo Houkes.

**Project administration:** Andrea Kis.

**Software:** Andrea Kis, Elena Mas Tur, Daniël Lakens.

**Supervision:** Elena Mas Tur, Daniël Lakens, Krist Vaesen, Wybo Houkes.

**Validation:** Elena Mas Tur, Daniël Lakens.

**Visualization:** Andrea Kis.

**Writing – original draft:** Andrea Kis.

**Writing – review & editing:** Andrea Kis, Elena Mas Tur, Daniël Lakens, Krist Vaesen, Wybo Houkes.

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
