## [Decision Letter · Decision Letter 0]

24 May 2022

PONE-D-22-05381Leaving academia: PhD attrition and unhealthy research environmentsPLOS ONE

Dear Dr. Kis,

Thank you for submitting your manuscript to PLOS ONE. After careful consideration, we feel that it has merit but does not fully meet PLOS ONE’s publication criteria as it currently stands. Therefore, we invite you to submit a revised version of the manuscript that addresses the points raised during the review process.

We look forward to receiving your revised manuscript.

Kind regards,

Alberto Baccini, Ph.D.

Academic Editor

PLOS ONE

Journal Requirements:

Reviewers' comments:

Reviewer's Responses to Questions

**Comments to the Author**

1. Is the manuscript technically sound, and do the data support the conclusions?

Reviewer #1: Yes

Reviewer #2: Yes

2. Has the statistical analysis been performed appropriately and rigorously? 

Reviewer #1: No

Reviewer #2: I Don't Know

3. Have the authors made all data underlying the findings in their manuscript fully available?

Reviewer #1: Yes

Reviewer #2: Yes

4. Is the manuscript presented in an intelligible fashion and written in standard English?

Reviewer #1: Yes

Reviewer #2: Yes

5. Review Comments to the Author

Reviewer #1: First and foremost, I apologize for my late review. Second, I also apologize for accepting this manuscript even though it does not lie in the center of my area of expertise. When I was approached for review, my initial reading seemed to imply a somewhat different kind of article than I eventually found. Nevertheless, being so late, I did my best.

I found this study to me timely, well-designed and its limitations well-formulated. I do not have any major concerns.

In the introduction, when referring to previous work, it may be warranted to emphasize the gaps in our knowledge that this current work is meant to address. Such a section could then be referenced in the discussion, especially given many of the found correlations were weak/moderate and discussed to which extent these gaps have been filled by the current study or which need further research. As it is written now, some readers may feel that much of what was asked in this study was already known, questioning the need for another study. Most readers likely will like to know why this study was done and whether the data collected answered the questions it was designed to answer.

Surprisingly, as one of the authors is also author on an article entitled "justify your alpha", I could not find the section in the M&M where the authors "transparently report and justify all choices they make when designing a study, including their alpha" value of 5%. With many valid reasons for setting the alpha value lower and given that one of the authors is a prominent critic of such reasons, such motivations and explanations appear warranted here. As many of the effects presented here seem to be significant at the .001 value, the authors could even adopt an alpha value of, say, 0.5%, likely without much consequence for their main conclusions. With many/most of their effects being weak/moderate, such considerations deserve special diligence.

Reviewer #2: This is an interesting and unique study of PhD candidates’ views of leaving academia. I have some suggestions for additions, changes, and clarifications:

• Mention the research methods used in the abstract and introduction.

• In the introduction, briefly acknowledge reasons for leaving other than those explored specifically in the study (e.g., lack of career prospects).

• Some of the sections overlap somewhat: e.g., 1. and 2.1 has some repetition; 3.1 describes context rather than methods.

• Express percentages consistently (e.g., with single, double, or no decimal points).

• Include footnote 1 in the main text, as it’s important to some of your later discussion.

There are also a few areas in the paper where the authors make assumptions that have been explored (and challenged) through previous research on leaving academia. For instance, regarding the desirability of ‘passion’ for academic research, which can negatively impact scholars, leading them to accept long-term precarious roles.

There is also the question of the desirability of low attrition (which is often presumed in studies of doctoral attrition, as well as by university managers, governments, and funding bodies). However, such scholarship ignores the fact that PhD programs have become more and more time limited (and poorly funded) in recent years. It also tends to avoid questions about the difficulty of PhD studies, and the role that research methods and resourcing play in influencing time to completion (e.g., the time commitment of intensive qualitative research, or the impact of being in a poorly funded discipline where supervisors are overworked).

Some research to cite includes:

• Barcan, R. 2019. Weighing up futures: experiences of giving up an academic career, in C. Manathunga and D. Bottrell (eds.), Resisting neoliberalism in higher education volume II: prising open the cracks, 43–64. Basingstoke: Palgrave Macmillan.

• Coin, F. 2017. ‘On quitting: the labour of academia’, Ephemera: Theory and Politics in Organization 17: 705–19.

• McKenzie, Lara 2021. ‘Unequal expressions: emotions and narratives of leaving and remaining in precarious academia’, Social Anthropology 29(2): 527–542.

6. PLOS authors have the option to publish the peer review history of their article (what does this mean?). If published, this will include your full peer review and any attached files.

Reviewer #1: **Yes: **Björn Brembs

Reviewer #2: **Yes: **Lara McKenzie

---

## [Author Response · Author response to Decision Letter 0]

8 Jul 2022

Dear Dr. Alberto Baccini,

Thank you for giving us the opportunity to submit a revised draft of our manuscript to PLOS ONE. We appreciate the time and effort that you and the reviewers have dedicated to providing your valuable feedback on our manuscript. We are grateful to the reviewers for their insightful comments and suggestions. We have been able to incorporate changes to reflect most of the suggestions provided by the reviewers. We have highlighted the changes within the manuscript.

Here is a point-by-point response to the reviewers’ comments and concerns:

Reviewer Comments to the Author

Reviewer #1: First and foremost, I apologize for my late review. Second, I also apologize for accepting this manuscript even though it does not lie in the center of my area of expertise. When I was approached for review, my initial reading seemed to imply a somewhat different kind of article than I eventually found. Nevertheless, being so late, I did my best.

I found this study to me timely, well-designed and its limitations well-formulated. I do not have any major concerns.

COMMENT 1.1:

In the introduction, when referring to previous work, it may be warranted to emphasize the gaps in our knowledge that this current work is meant to address. Such a section could then be referenced in the discussion, especially given many of the found correlations were weak/moderate and discussed to which extent these gaps have been filled by the current study or which need further research. As it is written now, some readers may feel that much of what was asked in this study was already known, questioning the need for another study. Most readers likely will like to know why this study was done and whether the data collected answered the questions it was designed to answer.

ANSWER 1.1:

We agree that we could have made more explicit what our paper contributes to the literature. We now write in the introduction (p. 5):

“The current study aims to find additional evidence for the importance of these different factors in explaining doctoral attrition. Additionally, given that the factors may be significantly correlated, we aim to contribute to the existing literature by examining these correlations and investigating the combined effects of: (1) the research climates in which PhD candidates operate; (2) their experiences of misconduct and questionable research practices; and (3) concerns related to supervisors’ behavior.”

We come back to this point in the “Discussion and Conclusion” section (p. 31)

COMMENT 1.2:

Surprisingly, as one of the authors is also author on an article entitled "justify your alpha", I could not find the section in the M&M where the authors "transparently report and justify all choices they make when designing a study, including their alpha" value of 5%. With many valid reasons for setting the alpha value lower and given that one of the authors is a prominent critic of such reasons, such motivations and explanations appear warranted here. As many of the effects presented here seem to be significant at the .001 value, the authors could even adopt an alpha value of, say, 0.5%, likely without much consequence for their main conclusions. With many/most of their effects being weak/moderate, such considerations deserve special diligence.

ANSWER 1.2:

We would like to thank the reviewer for reminding us of best practices. Our research is descriptive, and we did not test any hypotheses following the Neyman-Pearson philosophy (which underlies the arguments in Lakens et al. (2018) to 'Justify Your Alpha'). For example, we did not choose a sample size based on power, but based on feasibility, with the expectation that the sample size would be large enough to achieve accurate estimates. 

The descriptives reported in Figure 1 to 8 have no p-values, and we report all relevant correlations (without selection on significance) in Tables 1 to 8. Those tables are the only place where we indicate whether effects are statistically significant at the 0.05 and 0.001 levels (as the reviewer correctly observes, many of the correlations are significant at a very low alpha level). 

One could remove all references to alpha levels, and purely focus on descriptive statistics. However, this introduces the risk that personal biases lead to an overinterpretation of random variation as a signal when describing the data. We wanted to clearly communicate that when we describe an effect, there is a low probability we do so in error (i.e that the effect is actually 0). To transparently signal to readers that patterns in correlations we discuss are unlikely to be random noise, we have included asterisks that indicate statistical significance at the 0.05 and 0.001 levels. Of course, which correlations are statistically significant follow directly from the sample size (N = 391) so it is straightforward to choose any alpha level, and compute what the critical effect size is. We have now explicitly added this information, also for the alpha level of 0.005. 

The reviewer correctly points out the usefulness of being explicit about this statistical approach. Therefore, we have added the following: 

“We present descriptive results for all relevant survey questions in Figures, and provide Tables with all relevant correlations between survey questions. When interpreting the data, we discuss general patterns and highlight the most important variables (either in terms of frequencies, or in terms of the size of the correlation). We did not test any hypotheses. Given the sample size of N = 391 PhD students, correlations above 0.180, 0.210, 0.221, and 0.244 are statistically significant at the 0.05, 0.01, 0.005, and 0.001 level in a two-sided test, respectively. For the reader's convenience we flag correlations significant at alpha levels of 0.05 and 0.001 in all correlation tables to transparently communicate that there is a low probability we mistake random variation as a true effect.“

Reviewer #2: This is an interesting and unique study of PhD candidates’ views of leaving academia. I have some suggestions for additions, changes, and clarifications:

COMMENT 2.1:

• Mention the research methods used in the abstract and introduction.

• In the introduction, briefly acknowledge reasons for leaving other than those explored specifically in the study (e.g., lack of career prospects). 

• Some of the sections overlap somewhat: e.g., 1. and 2.1 has some repetition; 3.1 describes context rather than methods.

• Express percentages consistently (e.g., with single, double, or no decimal points).

• Include footnote 1 in the main text, as it’s important to some of your later discussion. 

ANSWER 2.1: 

We would like to thank the reviewer for outlining these specific details. Following these suggestions we made a range of changes to the text:

- We added the following lines describing the research methods we used in the abstract and the introduction: 

“We gathered quantitative self-report estimations of the perceptions of PhD candidates using an online survey tool and then conducted descriptive and within-subject correlation analysis of the results.”

“In the remainder of this paper, we first present the quantitative questions we used in our online survey tool. After discussing our results in terms of descriptive statistics, we then shift our focus on how PhD candidates’ experiences in their immediate research environment (i.e., the research climate where they work, questionable research practices, and supervisors’ conduct) relate to their considerations of leaving academia or quitting their PhD, as expressed by correlations”

- To decrease the number of overlaps, we decided to merge sections 1. Introduction and 2. Background literature together and deleted or restructured some of our text. We separated the context of the study from the literature and the methods parts, it is now a section by itself. We also removed the previous conclusion section, and adapted the discussion to “Discussion and Conclusion”. We trust that this new organization is more logical and clearer.

- We broadened our description of reasons for attrition and leaving:

“Many factors have been suggested as contributing to attrition and, more broadley, to intentions of leaving academia, including: funding, quality of supervision, scientific discipline, exposure to questionable research practices, institutional factors, organizational climate, involvement and socialization in the academic environment, community support, mental health, financial and nonfinancial costs, a lack of career prospects, and personal factors, such as life situations and attitudes towards doctoral studies (Arlinghaus & Kekecs, 2018; Bair & Haworth, 2004; Berdanier et al., 2020; Devos et al., 2017; Gardner, 2009; Golde, 1998; Groenvynck et al., Larson et al., 2014; 2013; Leijen et al., 2016; Mattijssen et al., 2020; Pyhältö et al., 2012; Skopek et al., 2020; Sverdlik et al., 2018; van de Schoot et al. 2013; van Rooij et al., 2019; Wollast et al., 2018).”

- We decided to express percentages consistently, with no decimal points throughout the paper. 

- We included all footnotes in the main text.

COMMENT 2.2:

There are also a few areas in the paper where the authors make assumptions that have been explored (and challenged) through previous research on leaving academia. For instance, regarding the desirability of ‘passion’ for academic research, which can negatively impact scholars, leading them to accept long-term precarious roles.

There is also the question of the desirability of low attrition (which is often presumed in studies of doctoral attrition, as well as by university managers, governments, and funding bodies). However, such scholarship ignores the fact that PhD programs have become more and more time limited (and poorly funded) in recent years. It also tends to avoid questions about the difficulty of PhD studies, and the role that research methods and resourcing play in influencing time to completion (e.g., the time commitment of intensive qualitative research, or the impact of being in a poorly funded discipline where supervisors are overworked).

ANSWER 2.2: 

We would like to thank the reviewer for this valuable suggestion. We agree that much of the literature assumes that attrition is undesirable, despite clear indications that PhD candidates may have good reasons to quit or leave. Still, the context of our study provides specific reasons for making the assumption that attrition is undesirable — although it is an assumption that is best made explicit. Thus, in response to this comment, we made a number of revisions. First, we now introduce the study in terms of factors “contributing to attrition and, more broadly, to intentions of leaving academia” rather than of “concerns”. Second, we make explicit and contextualize the undesirability assumption by adding the following to the introduction: 

Thus, most studies on PhD candidates who quit academia assume that this is undesirable and call for preventive measures; an assumption reflected by the value-laden term “attrition”. Some literature questions this assumption, and emphasizes that there may be good reasons at early career stages to quit or leave academia, such as the (increasingly) uncertain longer-term prospects and (decreasing) availability of job resources (see e.g., Coin 2017; McKenzie 2021). Although we acknowledge that low attrition is not necessarily desirable, our study focuses on factors connected to PhD students’ research climate and their exposure to questionable research practices and questionable professional conduct, including poor supervision. Lowering attrition due to these factors is part of a university’s duties of care.

COMMENT 2.3:

Some research to cite includes:

• Barcan, R. 2019. Weighing up futures: experiences of giving up an academic career, in C. Manathunga and D. Bottrell (eds.), Resisting neoliberalism in higher education volume II: prising open the cracks, 43–64. Basingstoke: Palgrave Macmillan.

• Coin, F. 2017. ‘On quitting: the labour of academia’, Ephemera: Theory and Politics in Organization 17: 705–19.

• McKenzie, Lara 2021. ‘Unequal expressions: emotions and narratives of leaving and remaining in precarious academia’, Social Anthropology 29(2): 527–542. (interviews with postgrads/precarious academics, so not PhDs)

ANSWER 2.3:

We thank you for your suggestions of citable sources. Some of these papers we included in our background readings before we started conducting the research, others we read now. All papers touch on the problems postgraduate academics face throughout their (often) precarious career paths so they helped us delineate the boundaries of our study. However, we decided not to include all of them in the final manuscript, because they fell out of scope with our purpose and target population, given the fact that in The Netherlands PhD students get a 4 year contract with a normal living wage, build pensions, receive a holiday allowance, are entitled to 16 weeks of pregnancy leave, and other basic labor rights. Those that we included were cited in discussing factors for quitting or leaving academia at early career stages (see Answer 2.2 above).

---

## [Decision Letter · Decision Letter 1]

8 Sep 2022

Leaving academia: PhD attrition and unhealthy research environments

PONE-D-22-05381R1

Dear Dr. Kis,

We’re pleased to inform you that your manuscript has been judged scientifically suitable for publication and will be formally accepted for publication once it meets all outstanding technical requirements.

Kind regards,

Alberto Baccini, Ph.D.

Academic Editor

PLOS ONE

Additional Editor Comments (optional):

Reviewers' comments:

Reviewer's Responses to Questions

**Comments to the Author**

1. If the authors have adequately addressed your comments raised in a previous round of review and you feel that this manuscript is now acceptable for publication, you may indicate that here to bypass the “Comments to the Author” section, enter your conflict of interest statement in the “Confidential to Editor” section, and submit your "Accept" recommendation.

Reviewer #1: All comments have been addressed

Reviewer #3: (No Response)

2. Is the manuscript technically sound, and do the data support the conclusions?

Reviewer #1: Yes

Reviewer #3: Yes

3. Has the statistical analysis been performed appropriately and rigorously? 

Reviewer #1: Yes

Reviewer #3: Yes

4. Have the authors made all data underlying the findings in their manuscript fully available?

Reviewer #1: Yes

Reviewer #3: Yes

5. Is the manuscript presented in an intelligible fashion and written in standard English?

Reviewer #1: Yes

Reviewer #3: Yes

6. Review Comments to the Author

Reviewer #1: The authors have more than adequately addressed all of my minor concerns.

As this form field requires additional characters, I have added this sentence.

Reviewer #3: Thank you for the opportunity to read this important paper exploring PhD student attrition rates. The sample size is a major strength and consideration of ethical and professional issues in the graduate research environment is particularly valuable. A good degree of context is provided for the unique situation of PhD students in The Netherlands that aids reader comprehension. It would be good to see some more explanation regarding why Likert scales were used and the limitations of statistical analyses for these scores. In the recommendations section it would also be good to note where current failings are occurring and what specific remediation policies are advised (e.g. there is acknowledgement that some institutions are already following some of the recommendations and yet they are clearly not working - why?). It would also be good to expand the conclusions section to make it more holistic, as the paper highlights many important points that are not included here.

7. PLOS authors have the option to publish the peer review history of their article (what does this mean?). If published, this will include your full peer review and any attached files.

Reviewer #1: **Yes: **Björn Brembs

Reviewer #3: **Yes: **Evie Kendal

---

## [Editor Report · Acceptance letter]

13 Sep 2022

PONE-D-22-05381R1 

Leaving academia: PhD attrition and unhealthy research environments 

Dear Dr. Kis:

I'm pleased to inform you that your manuscript has been deemed suitable for publication in PLOS ONE. Congratulations! Your manuscript is now with our production department. 

Kind regards, 

on behalf of

Prof. Alberto Baccini 

Academic Editor

PLOS ONE